# Structurally Dependent Electrochemical Properties of Ultrafine Superparamagnetic ‘Core/Shell’ γ-Fe_2_O_3_/Defective α-Fe_2_O_3_ Composites in Hybrid Supercapacitors

**DOI:** 10.3390/ma14226977

**Published:** 2021-11-18

**Authors:** Oleg Bazaluk, Andrii Hrubiak, Volodymyr Moklyak, Maria Moklyak, Lina Kieush, Bogdan Rachiy, Ivan Gasyuk, Yurii Yavorskyi, Andrii Koveria, Vasyl Lozynskyi, Serhii Fedorov

**Affiliations:** 1Belt and Road Initiative Institute for Chinese-European Studies (BRIICES), Guangdong University of Petrochemical Technology, Maoming 525000, China; bazaluk@ukr.net; 2G.V. Kurdyumov Institute for Metal Physics of the N.A.S. of Ukraine, 36 Academician Vernadsky Boulevard, 03142 Kyiv, Ukraine; mvvmcv@gmail.com; 3The Faculty of Physics and Technology, Vasyl Stefanyk Precarpathian National University, 56 Shevchenko Str., 76018 Ivano-Frankivsk, Ukraine; mariamoklyak@gmail.com (M.M.); bogdan_rachiy@ukr.net (B.R.); gasyukim@gmail.com (I.G.); 4National Metallurgical Academy of Ukraine, 4 Gagarin Av., 49600 Dnipro, Ukraine; linakeush@gmail.com (L.K.); fedorov.pte@gmail.com (S.F.); 5Department of Physical Materials Science and Heat Treatment, National Technical University of Ukraine “Igor Sikorsky Kyiv Polytechnic Institute”, 35 Politekhnichna Str., 03056 Kyiv, Ukraine; yaryra1990@gmail.com; 6Department of Chemistry, Dnipro University of Technology, 49005 Dnipro, Ukraine; Koverya.A.S@nmu.one; 7Department of Mining Engineering and Education, Dnipro University of Technology, 49005 Dnipro, Ukraine; lvg.nmu@gmail.com

**Keywords:** ultrafine composite, Mossbauer spectroscopy, phase composition, superparamagnetism, defective structure, capacity

## Abstract

The paper presents a method for obtaining electrochemically active ultrafine composites of iron oxides, superparamagnetic ‘core/shell’ γ-Fe_2_O_3_/defective α-Fe_2_O_3_, which involved modifying sol-gel citrate synthesis, hydrothermal treatment of the formed sol, and subsequent annealing of materials in the air. The synthesized materials’ phase composition, magnetic microstructure, and structural, morphological characteristics have been determined via X-ray analysis, Mossbauer spectroscopy, scanning electron microscopy (SEM), and adsorption porometry. The mechanisms of phase stability were analyzed, and the model was suggested as FeOOH → γ-Fe_2_O_3_ → α-Fe_2_O_3_. It was found that the presence of chelating agents in hydrothermal synthesis encapsulated the nucleus of the new phase in the reactor and interfered with the direct processes of recrystallization of the structure with the subsequent formation of the α-Fe_2_O_3_ crystalline phase. Additionally, the conductive properties of the synthesized materials were determined by impedance spectroscopy. The electrochemical activity of the synthesized materials was evaluated by the method of cyclic voltammetry using a three-electrode cell in a 3.5 M aqueous solution of KOH. For the ultrafine superparamagnetic ‘core/shell’ γ-Fe_2_O_3_/defective α-Fe_2_O composite with defective hematite structure and the presence of ultra-dispersed maghemite with particles in the superparamagnetic state was fixed increased electrochemical activity, and specific discharge capacity of the material is 177 F/g with a Coulomb efficiency of 85%. The prototypes of hybrid supercapacitor with work electrodes based on ultrafine composites superparamagnetic ‘core/shell’ γ-Fe_2_O_3_/defective α-Fe_2_O_3_ have a specific discharge capacity of 124 F/g with a Coulomb efficiency of 93% for current 10 mA.

## 1. Introduction

Iron oxide compounds continue to be one of the most important transition metal oxides of technological importance. The formation of nanostructured states of iron oxides along with standard long-known applications such as catalysts [1], pigments [2], coatings [3], and gas sensors [4] opens up the possibility of their use in the fields of magnetic information storage, medicine, wastewater treatment and storage, and conversion devices [5,6,7,8,9]. Usually, the effective use of the material, especially of ultrafine iron compounds, is determined by an adapted set of structural and morphological characteristics, phase composition, and physicochemical properties, the control of which should be carried out at the synthesis stage [10,11,12,13]. There are such parameters as temperature and pH of the reaction medium, concentration of ions in solution, nature of salt anion and cation of an alkaline agent, presence of uncontrolled or specially introduced impurities in solution, and reagent feed rate that are typical of synthesized by all known ‘wet soft chemical methods’.

One of the effective methods of ‘soft’ chemistry for obtaining ultrafine oxide phases with controlled composition, dispersion, particle size, and a given morphology is a hydrothermal treatment of solutions of iron salts [14,15,16]. The peculiarity of using the hydrothermal synthesis method is to increase the solubility of inorganic salts in water with increasing temperature and pressure and the subsequent sequential initialization of nonequilibrium competing processes of hydrolysis, nucleation, polycondensation, aggregation, which open up prospects for the formation of new defective material structures. The mechanisms of formation of oxide and hydroxide phases of iron from solutions under hydrothermal conditions will be influenced by the main factors: pH of the medium, the presence of a chelating agent, and the heat treatment temperature. In the case of neutral and alkaline media, the processes of hydrolysis and polycondensation of germs of the oxide phase (hematite) will prevail, which will cause the deposition of highly crystalline material. In an acidic environment, hydrolysis processes will compete with deprotonation, which brings to the fore the factor of processing time. After a short processing time, porous hematite is formed, which with increasing duration of hydrothermal conditions at temperatures of 150–250 °C, undergoes recrystallization and formation of more crystalline phases [17]. However, the clear answer to which parameters have a decisive influence on the phase and dispersed compositions of the formed ultrafine iron compounds remains, in particular the mechanism of action of the chelating agent, a matter of debate, which hinders the introduction of new technologies of reproducible synthesis of materials with optimal functional properties. In addition, the question of the formation of metastable structures is open, especially in post-processing.

This work aims to study the regularity of phase formation of ultrafine oxide compounds of iron obtained by hydrothermal treatment of iron citrate salt and the influence of further temperature annealing on phase transformation processes structural-morphological conductive characteristics and conductive characteristics, and the possibility of effective use of formed materials. 

## 2. Materials and Methods

A modified method was developed to form iron oxide compounds, including sol-gel citrate synthesis and hydrothermal treatment [10]. Firstly, a solution of iron citrate was formed, which provided the slow drip addition of 0.4 M aqueous solutions of iron nitrate (Fe(NO_3_)_3_·9H_2_O) to 0.4 M aqueous solution of citric acid (C_6_H_8_O_7_·H_2_O) with continuous stirring on a magnetic stirrer (“Ukrorgsintez”, Ukraine), at a temperature of 50 °C for 2 h. All materials were manufactured in Ukrreahim, Kyiv, Ukraine. The molar ratio of the original precursors was 1:1. The formed colloidal solution of iron citrate (pH = 0.3) was placed in a high-pressure hydrothermal reactor (“Ukrorgsintez”, Ukraine) and subjected to treatment at 120 °C for 5 h. As a result of treatment, an orange precipitate precipitated in the solution. The initial pH value of the solution after hydrothermal treatment was =1.5. The precipitate formed was washed with distilled water to a neutral pH = 7 aqueous suspension. Subsequently, the precipitate was dried in air at a temperature of 60 °C for 24 h. The result was source material for research (sample K1). The obtained powder was annealed at temperatures of 150 °C (sample K2), 250 °C (sample K3), and 350 °C (sample K4) in the air for 2 h with material sampling at each annealing temperature.

Studies of the synthesized materials’ phase composition and crystal structure were performed using a DRON-3.0 diffractometer (Instrument-making enterprise Burevestnik, Russia) and an Ultima IV diffractometer (Rigaku, Japan) with monochromatic radiation CuKα. Samples were taken on the device according to the Bragg-Brentano geometry using a Ni K_β_-filter (Rigaku, Japan). Qualitative analysis was performed via structural ICSD models and PowerCell software. Copper powder, annealed in a vacuum (850–900 °C for 4 h) with an average grain size of about 50 μm, was used as a reference sample to determine the instrumental expansion of the peak. The total width at half the maximum for the diffraction peak of the reference sample at 2θ = 43.38° was 0.129°. The size of the coherent scattering regions was calculated via the Scherrer equation:D = k·λ/β·cosθ,(1)
where k is the Scherrer constant (k = 0.9), λ is the wavelength (0.15405 nm), β is the maximum width at half height (in radians), and θ is the angular position of the maximum. A combination of Gaussian functions was used as the profile form.

To study the magnetic microstructure, the method of Mossbauer spectroscopy was used: MS-1104Em device (CJSC “CORDON”, Russia), constant acceleration mode, Co^57^ isotope in the chromium matrix, which serves as a source of γ-quanta, the line width of the metal α-Fe is 0.21 mm/s, and calibration isomeric shifts relative to α-Fe.

The surface area of the samples was measured by the Brunauer-Emmett-Teller (BET) method using a Quantachrome Autosorb device (Nova 2200e) (Quantachrome Instruments, USA) with nitrogen as the adsorbate. Adsorption isotherms were measured at 77 K. Studies of the morphology of the synthesized materials were performed using SEM JEOL JEM-100CX II (“Jeol” Japan).

The conduction properties of the materials were studied using an AUTOLAB PGSTAT12 (Metrohm Autolab, The Netherlands) impedance spectrometer with a FRA2 module. The surface area of the samples was measured by the BET method using a Quantachrome Autosorb device (Nova 2200e) with nitrogen as an adsorbate. Adsorption isotherms were measured at 77 K.

The study of the electrochemical properties of the synthesized materials was performed using the methods of cyclic voltammetry (CVA) and galvanostatic mode and the use of charge/discharge system TIONiT P2.00-xx (SPC “TIONiT”, UKRAINE). A study was performed using a three-electrode cell (“Ukrorgsintez,” Ukraine) to test the electrochemical activity of materials in the aqueous electrolyte. The working electrode was prepared by pressing into the nickel mesh a mixture of test material/acetylene carbon black (Super P)/polyvinylidene fluoride (n-methyl-2-pyrrolidone solution) in a mass ratio of 75:20:5 (%). A counter electrode used a platinum electrode, and the reference electrode was a chlorine-silver electrode. The electrolyte was a 3.5 M aqueous solution KOH. The performance of HSCs based on the synthesized materials was investigated in a two-electrode cell. As a negative (polarized) electrode used nanoporous carbon [18,19], the positive electrode was the synthesized material in each case. The electrolyte was a 3.5 M aqueous solution of KOH.

The initial composition for the negative and positive electrodes of the models of HSCs was a mixture of the test material and acetylene carbon black (Super P) ratio of 75:25. CVA and galvanostatic methods were used to analyze the efficiency of HSCs. The scanning range is 0–1 V. To estimate the specific capacity of the discharge (charge) of materials and HSCs on their basis in CVA and galvanostatic mode using the equation:C = q/ΔU·m,(2)
where q is the total charge accumulated during the discharge (charge), ΔU is the potential window used to scan the sample, m is the mass of the electrode mixture applied to the working electrodes.

## 3. Results

### 3.1. Structural and Morphological Characteristics

Figure 1 shows the X-ray diffraction patterns for synthesized samples annealed at different temperatures. The results of X-ray analysis showed that the source material, K1 is X-ray amorphous. Annealing at a temperature of 150 °C also does not cause the formation of crystalline phases because there are no clear maxima on the diffraction pattern. As for the starting material after annealing at 150 °C, the diffraction pattern of the K2 sample contains a halo in the region of small angles (20–30°), which indicates the presence of an X-ray-amorphous component in the investigated materials. The formation of the crystal structure begins after annealing at a temperature of 250–350 °C. In this case, the diffraction pattern of the K3 sample identifies the presence of crystalline phases of maghemite (γ-Fe_2_O_3_) and hematite (α-Fe_2_O_3_) with a predominance of the maghemite phase [20]. The phase ratio of the components is 78/22 = γ-Fe_2_O_3_/α-Fe_2_O_3_. The presence of expanded peaks indicates the low-crystalline state of the material in both phases. After annealing at 350 °C, the phase transformation process of the material γ-Fe_2_O_3_ → α-Fe_2_O_3_ continues, and the formation of more crystalline hematite (α-Fe_2_O_3_) becomes dominant in the material of 75%.

The Mossbauer spectra of the samples are presented in Figure 2. The Mossbauer spectrum of the source material K1 is a doublet line, which is uniquely identified due to the resonant absorption of γ-quanta on ^57^Fe nuclei in the paramagnetic state. The superposition of three doublet components was chosen to approximate the spectrum. Annealing of the material at 150 °C leads to a change in the magnetic microstructure. The Mossbauer spectrum of sample K2 is approximated by the superposition of two doublet components corresponding to ions Fe^3+^ and Fe^2+^. The ratio of the integral intensities of both components is Fe^3+^/Fe^2+^: 89/11.

For the K3 sample annealed at 250 °C, the Mossbauer spectrum is a superposition of the paramagnetic and magnetically ordered components. The ratio of the integrated intensities between sextuplet and doublet components corresponds to 64/46. Subsequent annealing of the material at a temperature of 350 °C (sample K4) causes a decrease in the doublet component of the Mossbauer spectrum, the parameters of which correspond to Fe^3+^ ions. As a consequence of the subsequent phase transformation during annealing of the material, namely the redistribution of sextuplet lines. The parameters of which Mossbauer spectra are summarized in Table 1.

Figure 3 shows structural and morphological features of the surface of the synthesized materials. There were evaluated by observation using scanning electron microscopy.

The initial precipitate after hydrothermal treatment consists of separated spherical particles with a size of approximately 1–2 μm. In addition, in the structure of the sample K1, cavities form through tunnel channels formed by combining adjacent particles. Annealing at 150 °C leads to partial sintering of particles in the zones of their direct contact—noticeable ‘bridges’ of sintering of neighboring particles. However, this process occurs locally, and in general, the size and shape of individual particles remain unchanged. As the annealing temperature increases, agglomeration processes occur between adjacent particles. For sample K3, the formation of continuous agglomerates of irregular shape with a size of about 2 μm is observed due to sintering and combining adjacent particles of weak crystalline material. Sample K4 is characterized by a continuous inhomogeneous surface, which arises from subsequent sintering and attachment of small particles to the existing crystallites. In addition, the formation of defective highly dispersed regions with a porous structure was recorded on their surface.

Additional information on the structural and morphological characteristics and the size of the specific surface area (SSA) of the synthesized materials were obtained by adsorption porometry. The recorded results showed that the material K1 after hydrothermal treatment does not have a crystalline surface structure. So, it was not possible to determine the value of the SSA for this material. A similar result was obtained for the material after annealing in air at a temperature of 150 °C. This is due to the X-ray amorphous state of these materials and the presence of structurally related water molecules. Therefore, the desorption of nitrogen will occur in parallel with the extraction adsorbed and structurally bound water from the material. The situation changes with increasing annealing temperature. Adsorption/desorption isotherms of samples K3 and K4 (Figure 4) have H4 type hysteresis according to the IUPAC classification, which is characteristic of mesoporous materials with pores with a diameter of 2–50 nm [21]. Characteristic of isotherms is the presence of high-pressure hysteresis, which is revealed in the divergence of the branches of adsorption and desorption in the region of high relative pressures. The reason for the formation of this hysteresis is associated with the phenomenon of polymolecular condensation in mesopores. Analysis of the desorption isotherm made it possible to determine the value of the SSA of the materials. It is established that the synthesized samples K3 and K4 are highly porous materials. A highly porous structure characterizes sample K3, with an SSA of 124 m^2^/h. Annealing at 350 °C causes the continuation of the phase transformation of the structure of γ-Fe_2_O_3_ → α-Fe_2_O_3_, which causes the particles to become more prominent, resulting in the SSA of the material decreasing 64 m^2^/g.

The low-crystalline state of the material and the presence of highly dispersed regions formed by superparamagnetic particles allow of K3 sample. The material is characterized by mesopores in the range of 3–6 nm and a local maximum around 4 nm (Figure 4, insert). In addition, there are micropores with a size of about 0.5 nm. The size distribution of pores for sample K4 is characterized by two maxima in the vicinity of mesopores measuring 4–6 nm and 7–8 nm (Figure 4, insert).

### 3.2. Electroconductive and Electrochemical Properties

To study the electrical properties of the synthesized samples, impedance analyses were carried out. It allowed analyzing the features of the frequency dependences of the electrical conductivity. The obtained experimental frequency dependences of the conductivity σ(ω) (Figure 5) are typical for semiconductor materials with a percolation conductivity mechanism, weak changes of the conductivity at low frequencies, and the increase with frequency enlarging [22].

The data obtained allowed us to conclude that the frequency-independent conductivity component σ*_dc_* for all samples varies in the range from ≈10^−6^ S/m to ≈10^−10^ S/m. The specific *dc* conductivity of K1 is about 7·10^−7^ S/m. Annealing at 150 °C causes sharp changes in the structure of the material and the valence state of iron ions, which also affects the electrical properties. There is a decrease in the specific conductivity σ*_dc_* of the sample K2 at the level of 10^−10^ S/m. For the source material in the low-frequency region, a frequency-independent σ*_dc_* region is observed. For sample K3, there is an increase in the frequency of the independent component of conductivity due to increases in percolation diffusion due to the immense value of the SSA of the defective structure of the material. Further sample annealing at 350 °C causes a partial increase in the frequency-independent component of the specific conductivity of the material by increasing the degree of crystallinity.

The study of electrochemical properties, particularly the identification of redox processes that contribute to the total capacity, turnover in aqueous electrolytes, and the magnitude of the electric charge of the synthesized iron oxides, was carried out using a three-electrode cell. Figure 6 presents the CVA of synthesized materials obtained at different temperatures at a scanning speed of 1.0 mV/s in the voltage range of −1.3–−0.3 V.

The specific values of charging and discharging capacities and Coulomb efficiency for synthesized materials calculated from CVA are presented in Table 2. The best capacitive characteristics are shown by the material obtained by annealing at a temperature of 250 °C. The specific discharge capacity is 177 F/g at a Coulomb efficiency of 85%. The worst capacity values are presented for the material K4, obtained by annealing at 350 °C.

Prototypes of HSCs were made based on the synthesized materials, and their electrochemical properties have been investigated. Cyclic voltammograms and galvanostatic charge/discharge curves for models of hybrid systems on synthesized materials at different annealing temperatures are shown in Figure 7, Figure 8, Figure 9 and Figure 10. The scanning speed varied in the range of 1–50 mV/s; the voltage range was 0–1 V. The peaks observed on the CVA curves are responsible for the redox reactions. Galvanostatic curves were obtained at load currents of 10–100 mA in the range of 0–1 V.

Table 3 and Table 4 summarize the models’ specific capacity characteristics HSCs on synthesized materials at different annealing temperatures obtained from cyclic voltammograms and galvanostatic charge/discharge curves.

As can be seen, the trends in the electrochemical activity of the synthesized materials in the three-electrode cell persist in the case of their use in HSCs. In particular, the maximum capacity values in galvanostatic and potentiodynamic modes are demonstrated by samples K2 and K3. HSCs based on sample K2 have a higher specific discharge capacity (134 F/g) in the galvanostatic mode at a current of 10 mA than sample K3 (124 F/g). Nevertheless, the Coulomb efficiency of 93% HSCs based on the K2 sample is lower at the same currents than the K3. At a current of 10 mA, the K2 sample has a Coulomb efficiency of 85%, while the K3 sample has 93%. In addition, with increasing discharge current, there is a tendency to increase the active resistance of the hydride system and reduce the capacity characteristics for all synthesis samples. However, for the HSCs based on the K3 sample, this trend is the smallest, and in the case of a current of 100 mA, the specific discharge capacity is 102 F/g with a Coulomb efficiency of 98%. In the CVA mode at low scanning speeds, the curves have a symmetrical rectangular shape for the HSCs based on the sample K3, indicating mainly the hybrid system’s capacitive mechanisms of charge accumulation [23]. The calculated specific discharge capacity for K3 in the CVA mode at a scanning speed of 1 mV/s is 126 F/g and a Coulomb efficiency of 80%. The obtained results make it possible to establish the efficiency of the ultrafine γ-Fe_2_O_3_/α-Fe_2_O_3_ composite of the K3 sample as the active component of the hybrid supercapacitor.

## 4. Discussion

### 4.1. Structural and Morphological Characteristics

From the analysis of diffractograms (Figure 1), it can be seen that the precipitate obtained after hydrothermal treatment and the material annealed at 150 °C are X-ray amorphous. The obtained result differs slightly from the data obtained earlier on the hydrothermal treatment of solutions of iron-containing salts. In particular, it is known that the hydrothermal treatment of solutions of iron nitrate in the temperature range 130–250 °C initiates the processes of high-temperature hydrolysis, which leads to the formation of crystalline hematite [17]. The concentration of precursors mainly determined the size, shape, and morphology of the material particles. Only dense vitreous iron oxide particles were observed at low concentrations of iron nitrate (0.01–0.05 M) in the final product. Increasing the concentration to 0.1–0.5 M leads to the formation of porous particles (40–100 nm) containing pores in the range of 5–20 nm. Further increase in concentration leads to the formation of a significant amount of nitric acid, which causes recrystallization and growth of hematite particles to hundreds of nanometers in the case of 4 M solution of iron nitrate. The authors propose a mechanism for the formation of mesoporous hematite particles due to concurrence in the nucleation of iron hydroxide and recrystallization of primary grain aggregates during hydrothermal treatment with the formation of the hematite phase.

In our case, due to the addition of citric acid to the nitrate solution, the mechanism of nucleation processes will be slightly different. The use of citric acid as a chelating agent shifts the equilibrium of the reaction of hydrolysis/deprotonation in the direction of deprotonation and the formation of stable hydro-complexes of iron citrate. In addition, the additional adsorption of these molecules on the surface of the formed hydro complexes prevents their aggregation. Most early studies claim that in neutral environments, formed hydro complexes of iron citrate are in a stoichiometric ratio of 1:1 [24], and the degree of hydrolysis for the charge of such a complex may vary [25,26]. The formation of hydro complexes of FeL, Fe_2_L_2,_ and Fe_3_L_3_ at close molar ratios of 1:1 of Fe^3+^ ions and citric acid anions in solution was also observed in Ref. [27]. The same authors claim that in an acidic environment, citrate complexes with a stoichiometric ratio of 1:1 also remain stable. Increasing the pH of solutions to alkaline values and increasing the relative content of anions causes an increase in hydrolysis processes and initiates polycondensation of coarsely dispersed polymers of iron hydroxide [27,28,29]. Thus, in an acidic environment and while maintaining the stoichiometric ratio of precursors 1:1, which is observed in our experiment, stable hydro complexes of iron citrate are formed, for which under normal conditions, no significant aggregation is observed. In the hydrothermal treatment of such a solution, the hydrolysis processes trigger the mechanism of new phase nucleation. However, the presence of anionic chelating agents does not allow to complete of the processes of polycondensation of iron hydroxide and its recrystallization into oxide phases by precipitation of crystalline iron oxide *x*Fe_2_O_3_·H_2_O [25]. An amorphous hydrogel of variable composition is formed, which was recorded by the X-ray diffraction method. Citric acid anions encapsulate the primary nuclei of iron hydroxide, resulting in the formation of small polymer chains, the deposition of which causes the formation of ultrafine material of sample K1. Further annealing at a temperature of 150 °C does not cause recrystallization of the material and the formation of crystalline phases. At this temperature, only the processes of extraction of structurally bound water and combustion of the organic component are initiated, which will affect the change of the magnetic microstructure and electrically conductive properties, as will be shown below.

During annealing at 250 °C, there is a phase transition and crystallization of fine oxide phases of iron, namely maghemite and hematite. Moreover, if the hematite phase is uniquely identified, the presence of maghemite is more challenging to establish. However, another structure of the material—magnetite (Fe_3_O_4_) is not excluded, as the X-ray structural separation of these phases is difficult. This is especially difficult to do for ultrafine samples due to the identity of the diffractograms of the defective spinel γ-Fe_2_O_3_ and the inverted Fe_3_O_4_, which differ only in the appearance of weak additional reflexes (111), (222), and (511) for Fe_3_O_4_ due to the ordering of the structure [30]. The calculated average size of the coherent scattering (ACR) regions for the γ-Fe_2_O_3_ phase is 6.0 ± 1.0 nm, and for the α-Fe_2_O_3_ phase is 10.0 ± 1.0 nm. The particle size is determined by measuring the width at half the height of the reflexes (220), (311), (400), (422), and (511) (Figure 1).

The K4 sample annealed at 350 °C is characterized by the preservation of the ultradisperse state of the γ-Fe_2_O_3_ phase and the formation of a more crystalline α-Fe_2_O_3_, the dominant component of which indicates the continuation of the transformation of the structure of γ-Fe_2_O_3_ → α-Fe_2_O_3_. The maghemite (25%) remains in a weakly crystalline state while maintaining the size of the ACR at about 6 nm. The average ACD size for α-Fe_2_O_3_ was calculated based on the determination of the width at half the height of the reflexes (012), (104), (110), (113), (024), and (116). The obtained ACR sizes for the γ-Fe_2_O_3_ phase are 16.0 ± 1.0 nm. 

It is the presence of phase transformation at an annealing temperature of 350 °C that suggested the formation of the γ-Fe_2_O_3_ phase at 250 °C since the transition of γ-Fe_2_O_3_ → α-Fe_2_O_3_ during annealing in the air is more probable than direct Fe_3_O_4_→α-Fe_2_O_3_ [7]. In Ref. [31], the transition of Fe_3_O_4_ → α-Fe_2_O_3_ during annealing is accompanied by the intermediate phase γ-Fe_2_O_3_ due to oxidation of the defective structure of the spinel in a reasonably wide temperature range, which in our case was not noticed. In addition, crystallographically isomorphic maghemite and magnetite differ in the presence of Fe^2+^ ions in the latter, which can be detected by Mossbauer spectroscopy. Therefore, for unambiguous identification of the phase composition of the synthesized samples of iron oxides and evaluation of their magnetic microstructure, appropriate Mossbauer studies were performed at room temperature (Figure 2).

For sample K1, the Mossbauer spectrum of which is a doublet line characteristic of the paramagnetic state of ions Fe^3+^, the fixed values of isomeric shift (δ) and quadrupole splitting (Δ) are consistent with the parameters for the phase Fe(OH)_3_ [32]. Based on these data and X-ray diffraction results, it can be concluded that due to the hydrothermal treatment of the iron citrate solution, there are processes of polycondensation of iron citrate complexes (their deprotonation), which leads to precipitation of iron oxide hydrogel of variable composition [33]. Drying this hydrogel partially removes adsorbed and structurally bound water and an amorphous structure Fe(OH)_3_. 

Annealing of xerogel Fe(OH)_3_ at 150 °C causes a change in the magnetic microstructure of the material along with Fe^3+^ ions on the Mossbauer spectrum of the sample K2 fixed component corresponding to the ions Fe^2+^. The parameters of the doublet component corresponding to the absorption of γ-quanta on ^57^Fe nuclei remain the same as for sample K1 [32]. Analysis of the doublet component parameters of the spectrum K2 corresponding to Fe^2+^ ions allowed us to establish the probable phase of this component. According to the calibration δ and Δ, Fe^2+^ ions are in a tetrahedral environment in the structure of amorphous Fe(OH)_2_. The obtained result can be explained as follows: with increasing annealing temperature, the dehydration of the material continues. Moreover, this process occurs spatially inhomogeneously, the regions where the extraction of structurally bound water coexisted with the regions of Fe(OH)_3_ [34]. In addition, the presence of structurally adsorbed carbon groups on the defective surface of the material is not excluded. In particular, (COO−) or C=C, which in the process of annealing and desorption capture oxygen vacancies, leading to irreversible reduction of iron ions Fe^3+^ → Fe^2+^. As a result of such transformations, a composite material of X-ray amorphous iron hydroxides Fe(OH)_3_/Fe(OH)_2_ was formed. The ratio of components will be determined by the defect of the surface structure and the presence of adsorbed surface groups. The authors recorded this type of effect of rearrangement of iron ions in various microporous and mesoporous ferrisilicates during redox processes [35].

Alternatively, organic components in the hydrogel of iron oxide after hydrothermal treatment should be kept in mind. In particular, iron citrate pentahydrate in the original sample K1 is not excluded, as the recorded parameters of one of the doublet, components correspond to the literature data for C_6_H_5_O_7_Fe·5H_2_O [36,37]. In this case, at a temperature of 150 °C, the presence of Fe^2+^ ions in the material can be explained by the transition of iron (III) pentahydrate to iron (II) oxalate during annealing (Equation (3)):(3)$$C_{6}H_{5}O_{7}\mathrm{Fe}\cdot 5H_{2}O\overset{anneling}{\to }{\mathrm{FeC}}_{2}O_{4}\cdot 2H_{2}O,$$

The analysis of the Mossbauer spectra of the K3 sample made it possible to confirm this material’s phase composition. Maghemite and magnetite are ferromagnets in which iron ions are in different tetrahedral (A) and octahedral (B) positions [7], which in turn corresponds to different sextets on the Mossbauer spectrum. In this case, magnetite is characterized by the presence of Fe^2+^ ions, and for each position of the ion, the magnitude of the effective magnetic field will be different. Therefore, the Mossbauer spectrum of magnetite is approximated by three sextet lines characterized by quadrupole splitting near zero and the value of the isomeric shift at the level 0.66 mm/s [32].

On the other hand, the maghemite has two sextet lines with relative values of isomeric shift (0.27 mm/s) and zero quadrupole splitting. Thus, analyzing the magnetically ordered component of the K3 spectrum, the presence of two broader sextuplet lines due to the resonant absorption of γ-quanta by ^57^Fe nuclei in γ-Fe_2_O_3_ particles was established. The third sextuplet line is uniquely identified as ^57^Fe nuclei in the hematite with defective structure, as evidenced by the reduced value of the effective magnetic field on the nucleus at the level of 510 kOe. Moreover, the ratio of the integrated intensities of the sextuplets corresponding to the phases of maghemite and hematite is 43/11, which is consistent with the results of X-ray phase analysis (X-ray diffraction), according to which a similar phase ratio was obtained.

The doublet component is 46% of the total integral intensity of the Mossbauer spectrum, and its parameters are identified as iron ions Fe^3+^. Given the weakly crystalline state of the maghemite of sample K3, the paramagnetic component in the spectrum can be explained by the development of the phenomenon of superparamagnetism for iron oxide particles [34,38]. There are literature data on the sizes of nanoparticles for which the transition to the superparamagnetic state occurs at room temperature. This is due to the difficulties in synthesizing dimensionally homogeneous materials, but it can be confirmed that the critical value is in the range of 10 nm [30]. Approximation of the paramagnetic component of the spectrum of sample K3 was presented as a superposition of two doublet components with similar values of isomeric shift values δ = 0.33 ± 0.05 mm/s and different values of quadrupole splitting Δ = 1.02 and Δ = 0.67 mm/s. This is due to the difference in the immediate environment of Fe^57^ nuclei, which are in crystal-nonequivalent positions [39,40,41]. For the K3 sample, the superparamagnetic state of the maghemite particles was recorded, and there are two doublet components of the specter corresponding to ^57^Fe nuclei with different near environments in these particles. The first doublet component with smaller quadrupole fission (D = 0.67 mm/s) indicates ^57^Fe short-range nuclei in superparamagnetic particles. While the second component with a large quadrupole split (D = 1.02 mm/s) indicates ^57^Fe nuclei in a highly ordered near environment. A highly defective material state characterized the last component with large quadrupole fission with an ordered environment of ^57^Fe nuclei.

Furthermore, since the ratio of the integrated intensities of these components is 9:37 towards the component with a larger quadrupole, it can be argued that the ^57^Fe nuclei with a disordered environment are in the near-surface region of superparamagnetic particles [42,43]. As an option, we can consider the formation of the structure ‘core-shell,’ for which there are non-equivalent positions of Fe^3+^ ions in the volume of γ-Fe_2_O_3_ nanoparticles in the superparamagnetic state and on their defective surface, which corresponds to a near asymmetric environment and is expressed by changing quadrupole. So, the following model of iron oxide phases formation due to the annealing of the sample K2 can be suggested. Heat treatment of composite X-ray amorphous iron hydroxide Fe(OH)_3_/Fe(OH)_2_ in the air causes its oxidation and subsequent removal of structurally bound water and the formation of weakly crystalline superparamagnetic ‘core-shell’ γ-Fe_2_O_3_ [39,44]. Particle agglomeration and phase transformation of the crystal structure of γ-Fe_2_O_3_→α-Fe_2_O_3_ take place in parallel. Thus, superparamagnetic ‘core-shell’ weakly crystalline γ-Fe_2_O_3_ particles, crystalline γ-Fe_2_O_3_ and defective low-crystalline α-Fe_2_O_3_ are present simultaneously in the material of sample K3.

In the Mossbauer spectrum of K4, as in the case of sample K3, there is a sextet corresponding to the phase of maghemite with iron nuclei in a magnetically ordered state of 34%. Moreover, it is approximated by two lines with different quadrupole splits (Table 1), corresponding to different environments of iron nuclei. The integral intensity of the sextet component corresponding to the magnetically ordered hematite increases up to 40% compared to the sample K3 (11%). Additionally, there are two sextuplet lines with different values of effective magnetic field and quadrupole splitting, corresponding to crystalline α-Fe_2_O_3_ with an effective field of 513 kE and α-Fe_2_O_3_ with a defective structure and an effective magnetic field—501 kE [40]. The available doublet component does not exceed 9% of the total integral intensity. As in the case of sample K3, it corresponds to iron nuclei in the superparamagnetic state of iron oxide particles. Based on cross-analysis of X-ray diffraction results and Mossbauer spectroscopy, the following annealing mechanism is proposed. When the temperature reaches 350 °C, annealing and aggregation of superparamagnetic γ-Fe_2_O_3_ particles are observed. At the same time, weakly crystalline particles of γ-Fe_2_O_3_ in the magnetically ordered state are transformed into the structure of hematite. Direct adhesion of superparamagnetic particles of maghemite to the surface of hematite crystallites with simultaneous phase transition is also probable. In this case, both processes are nonequilibrium, which is reflected in the formation of a defective structure of hematite, recorded by Mossbauer spectroscopy. The transition state of phase transformations is indicated by a relaxation component of the Mossbauer spectrum, the contribution of which is about 17%. Thus, annealing at 350 °C forms a defective composite γ-Fe_2_O_3_/α-Fe_2_O_3_.

The results of direct observations (Figure 3) can be explained considering the data of Mossbauer spectroscopy and the single-phase K3 sample according to X-ray diffraction. Since ^57^Fe nuclei in the crystalline states of the ultrafine ‘core/shell’ superparamagnetic γ-Fe_2_O_3_/defective α-Fe_2_O_3_ composite interact magnetically with the surrounding field, the magnetically ordered component of the Mossbauer spectrum corresponds to Fe^3+^ ions in the structure of the formed crystallites of approximately 2 μm in size. The inhomogeneity of the crystallite surface and the presence of structural defects probably causes the formation of areas of near-surface defect layer [39], in which iron ions due to the asymmetric environment and the presence of broken bonds are in the paramagnetic state and are characterized by high quadrupole splitting. The second paramagnetic component of the doublet component of the Mossbauer spectrum of K3, which corresponds to ^57^Fe nuclei in the superparamagnetic state, is probably due to the formation of porous, highly dispersed regions on the crystal surface γ-Fe_2_O_3_ superparamagnetic particles.

For the K4 sample was continues the simultaneous transformation of the crystal structure of the latter was continued according to the scheme γ-Fe_2_O_3_ → α-Fe_2_O_3_. This is confirmed by the growth of the calculated BET sizes from the diffraction pattern and the magnitude of the magnetically ordered component in the integral intensity of the Mossbauer spectrum of the K4 sample. Firstly, there is a general transformation of the material structure throughout the entire volume. Then the attachment of small particles of hematite to large crystallites occurs. This process at 350 °C is not complete, as evidenced by the presence of small particles on the surface of the crystals, and it is the ^57^Fe nuclei that can match the paramagnetic component present in the Mossbauer spectrum K4 sample.

### 4.2. Electroconductive and Electrochemical Properties

The approximation of the dependencies σ(ω) was calculated using the John Sher equation [45]:(4)$$\sigma \left(\omega \right)=\sigma_{dc}\left[1+{\left(\frac{\omega }{\omega_{h}}\right)}^{s}\right],$$
where σ_*dc*_—conductivity in the DC mode, ω_h_—the frequency of jumps of charge carriers, s—an indicator that characterizes the deviation of the system from the properties predicted by the Debye model and is a measure of interparticle interaction (0 < s < 1).

The parameters in all cases are in the vicinity of the value ≈1.0, which indicates the equally probable nature of multipositional jumps of charge carriers due to the increase in the degree of homogeneity of the studied systems. These samples are in a state close to the Debye model, and observed conductivity behavior could be explained by a small polaron hopping mechanism [46,47]. For the sample K2, increasing frequency becomes linearly dependent (Figure 5) due to probably complex percolation diffusion of electrons and the defective X-ray structure amorphous material and effects associated with the contribution of impurity conductivity. In addition, as the annealing temperature increases (samples K3 and K4), the characteristic value of the equilibrium conductivity to which the dependence σ(ω) goes increases with increasing frequency. At the same time, saturation is achieved at relatively higher frequencies, which is also due to the structural ordering of the material. Dependencies of this type are characteristic of the case of conductivity of materials arranged at the microscopic level.

Information on the kinetics of charge accumulation processes and the electrochemical activity of the synthesized materials in the proton electrolyte was obtained by cyclic voltammetry in a three-electrode cell (Figure 6). CVA of the initial material has an asymmetric appearance with clear redox maxima on the charging and discharging branches, indicating that the capacitance is induced mainly by Faraday processes. 

A reduction peak at a voltage of −1.20 V on the discharge branch and the corresponding oxidation peak on the charging branch at −0.95 V indicate the reversibility of the material’s redox processes and pseudocapacitive properties. On the anode branch, there are also slight widened maxima of −0.77 and −0.60 V. A sharp increase in current at the lower voltage limit is associated with the release of hydrogen on the surface of the electrode [48]. In this range, the material behaves as a non-polar electrode. 

The CVA curve obtained for the annealed material at 150 °C is also characterized by an asymmetric appearance and the presence on the cathode branch of a weakly expressed maximum in the vicinity of potential values of −1.20 V and a clear widened anode peak in the vicinity of −0.60 V. There are also maxima at −0.80 V and −0.10 V. The peak currents’ magnitude is larger than the initial material K1, which indicates a more significant contribution of the pseudo-capacitive accumulation processes for this material [49]. The absence of an apparent reduction peak may indicate a deterioration in the reversible properties of the material synthesized at a temperature of 150 °C. In general, comparing the CVA of samples K1 and K2, the similarity of the general appearance of the curves can be noticed, however, with the redistribution of peak current intensities on the anode branch between the potentials −0.95 and −0.60 V. The reason for this may be related to the course of rapid oxidative reactions involving Fe^2+^ ions on the surface of the electrode material and electrolyte anions, which is observed for sample K2 (for sample K1 on the CVA curve, only traces of maximum formation at the same potential).

For the sample K3 obtained by annealing at a temperature of 250 °C, the voltammetry has a symmetrical appearance about the potential axis with two distinct peaks corresponding to the maximum rate of oxidation and reduction processes characteristic of pseudocapacitive charge accumulation (Figure 6). The anode peak corresponds to a voltage of −0.70 V, the cathode of −1.10 V. These peaks represent rapidly reversible redox processes occurring at the interface between the composite phases on the one hand and the electrolyte on the other. The difference between the cathode and anode peak is 0.44 V, corresponding to the Faraday reversible reactions on the material [50]. The increase in the CVA area of the curve for sample K3 in comparison with K1 and K2 indicates the accumulation of a larger amount of charge on its surface, which is mainly provided by the course of Faraday pseudocapacitive reactions. 

There are no distinct peaks on the voltammogram curve of the material obtained by annealing at 350 °C (Figure 6). The formation of the capacity in this material is mainly due to the formation of an electrical double layer at the interface. In addition, the CVA curves show a sharp irreversible increase in current at the lower voltage limit due to the release of hydrogen on the surface of the electrode. There are no processes of pseudocapacitive accumulation of charge, despite the porosity of the material and the presence of mesopores. In all other samples, the formation of capacity occurs due to the combination and simultaneous flow of mechanisms of electrosorption with an electrical double layer and Faraday processes associated with the course of redox reactions. The difference in the capacitive characteristics of the synthesized materials is due to the different phase composition of the material, each of which is characterized by its efficiency in the proton electrolyte, which does not allow comparing the obtained data directly.

In the case of the prototype of HSC based on synthesis materials for the initial material charge/discharge, data curves were taken at 10, 20, and 30 mA current to confirm their pseudocapacitive behavior. In particular, plateaus are observed at discharge currents of 10 and 20 mA on the discharge curves, which are responsible for electrochemical adsorption/desorption or the course of redox reactions [51], confirmed by the course of CVA. 

From the analysis of CVA curves, we see that the formation of an electrical double layer makes the main contribution to the capacity of the prototype HSC. Moreover, voltammograms have a good rectangular shape for CVA curve models based on materials annealed at 150 and 250 °C.

As the discharge current increases, the specific capacitance decreases sharply (Table 3). On the one hand, this is due to the carbon material, in which the ohmic resistance increases with increasing discharge current due to the presence of micropores [52]. These micropores restrict access to the inner surface of the material. On the other hand, with the irreversibility of redox reactions at high discharge currents.

For sample K4 obtained at 350 °C and in which the dominant phase is crystalline hematite, the capacitive performance of prototypes of HSCs decreases sharply, although the electrical conductivity at a constant current for this material is higher than sample K3. Therefore, it can be confirmed that the predominant role of morphology, defects in the structure, and phase composition in developing electrochemical properties of synthesized materials.

The model based on the annealed material at a temperature of 250 °C has the best capacitive characteristics. This is primarily because the accumulation of charge in such a system occurs due to the reaction of cations with an electroactive material, followed by redox reaction and stable reversibility of pseudocapacitive reactions. The results obtained for sample K3 are due to the good structural and morphological characteristics of this ultrafine ‘core/shell’ superparamagnetic γ-Fe_2_O_3_/defective α-Fe_2_O_3_ composite. The presence of a defective hematite structure, ultra-dispersed maghemite, particularly in the superparamagnetic state, the presence of a developed mesoporous structure with high values of SSA in combination with good conductive properties provide good electron-ion transport of charges with the possibility of fast reactions. This structural and morphological characteristics level was achieved due to controlled hydrothermal synthesis followed by thermal annealing and the presence in the synthesis of a chelating agent, which encapsulated new phase nuclei in the reactor and prevented direct recrystallization of the structure with subsequent α-Fe_2_O_3_ crystal formation. 

## 5. Conclusions

The method for synthesizing ultrafine superparamagnetic ‘core/shell’ γ-Fe_2_O_3_/defective α-Fe_2_O_3_ composite with properties adapted for use in electrochemical capacitors has been developed. The phase composition and magnetic microstructure of iron hydroxides obtained by annealing Fe(OH)_3_ in the range of 150–350 °C were determined, and the mechanisms of phase transformations were analyzed. It was established that after annealing at 250–350 °C, there is the aggregation of superparamagnetic γ-Fe_2_O_3_ particles are observed. At the same time, weakly crystalline particles of γ-Fe_2_O_3_ in the magnetically ordered state are transformed into the structure of hematite. Direct adhesion of superparamagnetic particles of maghemite to the surface of hematite crystallites with simultaneous phase transition, which is reflected in the formation of a defective structure of hematite is also probable.

It was also established that the percolation diffusion of electrons determines the conductivity mechanism of synthesized iron hydroxide along with the defective structure and the contribution of the impurity conductivity of the surface adsorbed groups. The frequency-independent component of the conductivity σ*_d_* for all samples varies from ≈10^−6^ S/m to ≈10^−10^ S/m.

It was found that the formation of the capacitance is associated with the parallel course of pseudocapacitive charge accumulation due to the course of Faraday reactions and charge accumulation processes on the electrical double layer. In the proton electrolyte was optimized the hybrid electrochemical system and achieved maximum values of specific capacity of 177 with a Coulomb efficiency of 85% for the ultrafine ‘core/shell’ superparamagnetic γ-Fe_2_O_3_/defective α-Fe_2_O_3_ composite.

When testing hybrid supercapacitor based on composites at a constant current discharge in the voltage range of 0–1 V, a positive correlation was noticed between the structural and morphological characteristics (particularly the specific surface area), phase composition, and the operating values parameters of the corresponding prototypes of hybrid supercapacitor. The prototypes of hybrid-electric capacitors with work electrodes based on ultrafine composites superparamagnetic ‘core/shell’ γ-Fe_2_O_3_/defective α-Fe_2_O_3_ have a specific discharge capacity of 124 F/g with a Coulomb efficiency of 93% for current 10mA. The obtained results for ultrafine ‘core/shell’ superparamagnetic γ-Fe_2_O_3_/defective α-Fe_2_O_3_ composite are due to the successful structural and morphological characteristics of this composite, in particular the presence of defective hematite structure, the presence of ultra-dispersed maghemite, in particular in the superparamagnetic state, the presence of a developed mesoporous structure with high values of the specific surface area in combination with good conductive properties. The hybrid supercapacitor combines the capacitive accumulation of charge due to an electrical double layer formation and the course of pseudo-capacitive reactions.

## Figures and Tables

**Figure 1 materials-14-06977-f001:**
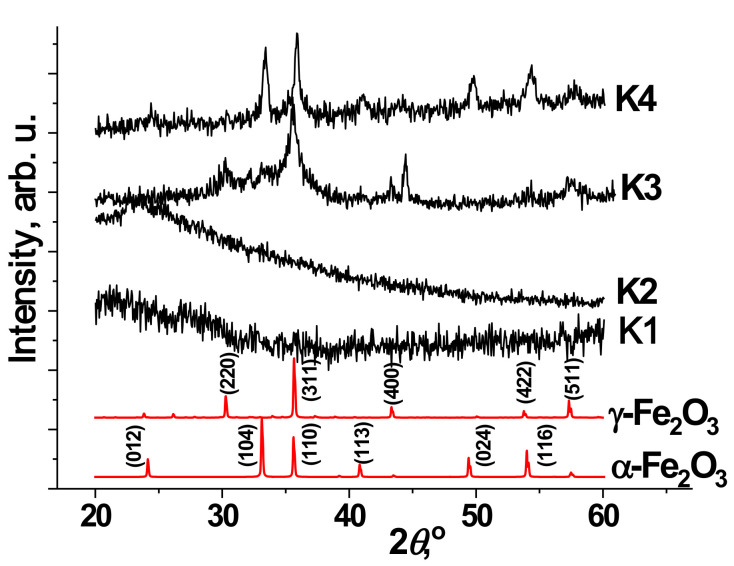
Diffractograms of synthesized material, after annealing at different temperatures and the model γ-Fe_2_O_3_ (JCPDS cards 39-1346) and α-Fe_2_O_3_ (JCPDS card 87-1164).

**Figure 2 materials-14-06977-f002:**
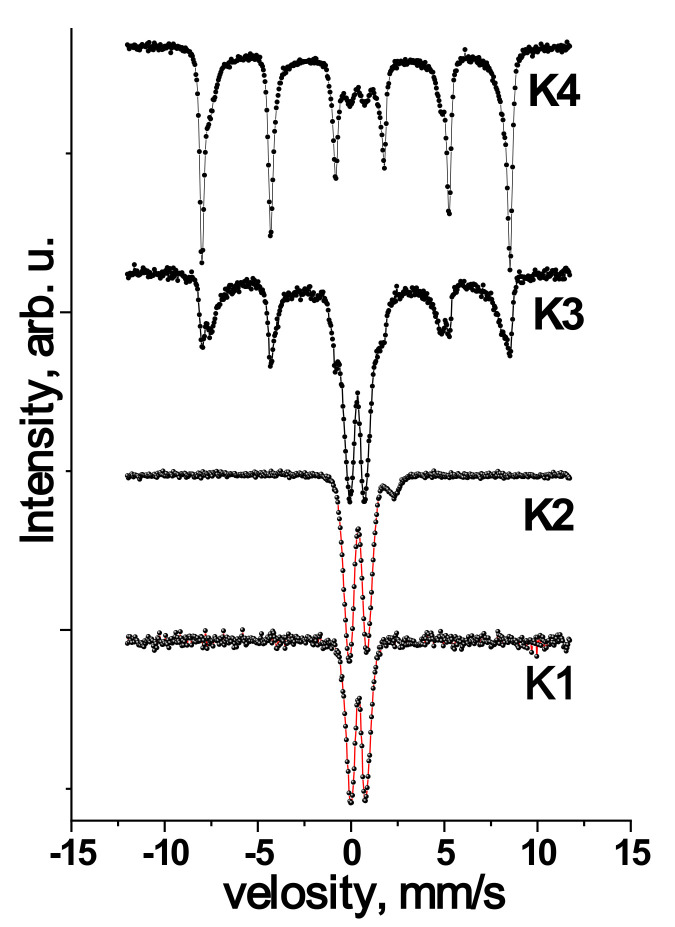
Mossbauer spectra of synthesized materials and after annealing at different temperatures.

**Figure 3 materials-14-06977-f003:**
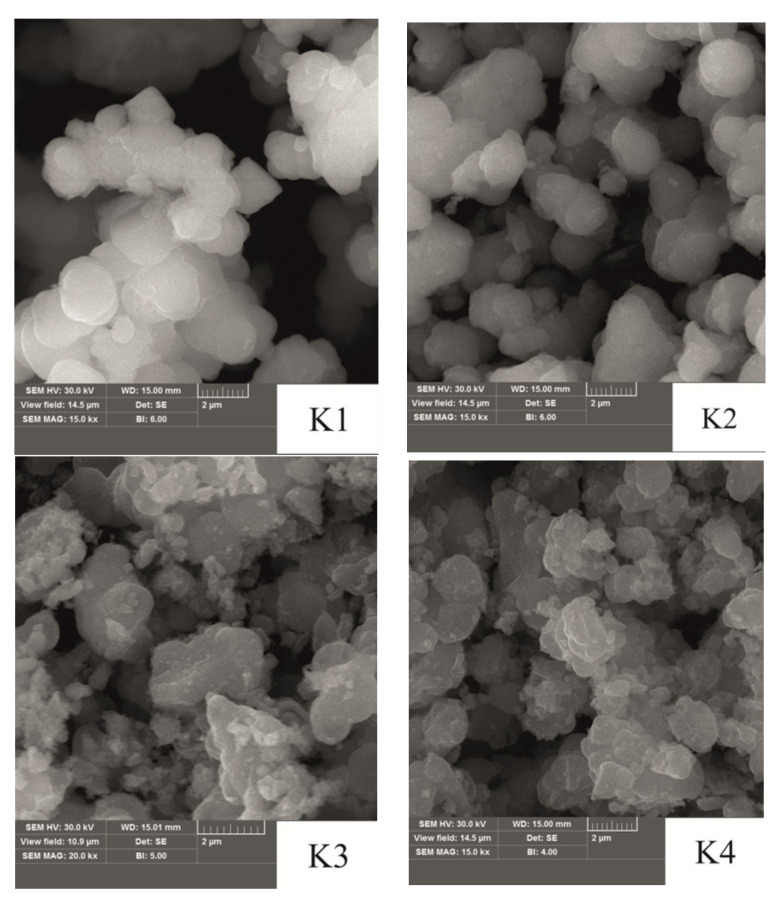
SEM images for the synthesized initial material K1 and obtained at annealing temperatures of 150 (K2), 250 (K3), and 350 °C (K4).

**Figure 4 materials-14-06977-f004:**
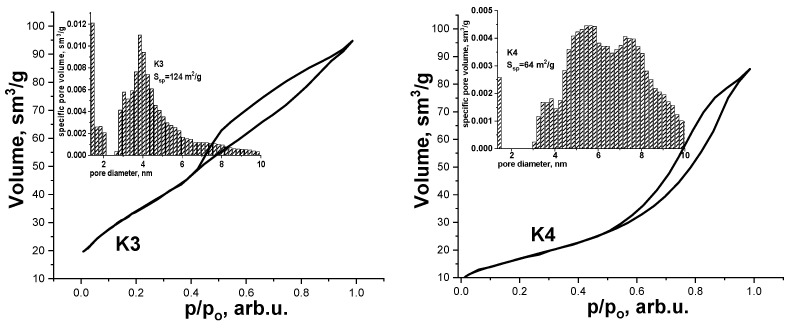
Adsorption-desorption isotherms for samples K3 and K4, the inserts show the corresponding size distribution of pores for synthesized materials.

**Figure 5 materials-14-06977-f005:**
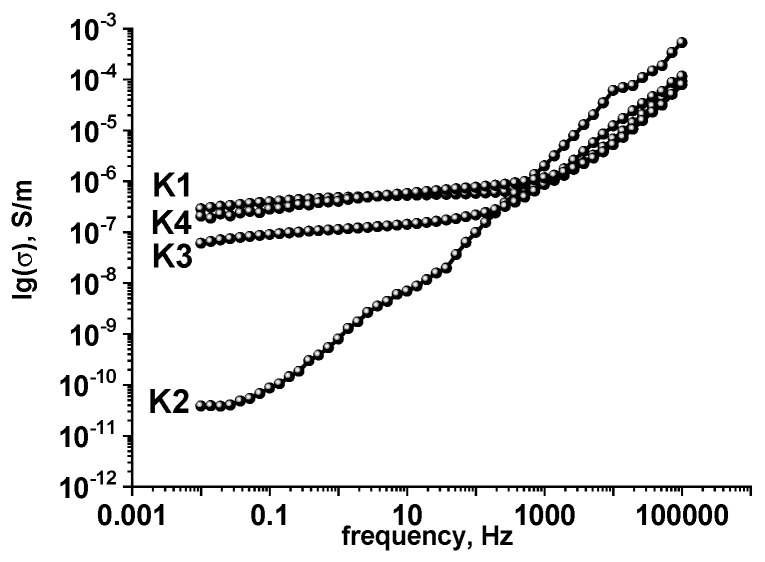
Frequency dependences of conductivities of synthesized materials obtained at different temperatures for samples K1, K2, K3, and K4.

**Figure 6 materials-14-06977-f006:**
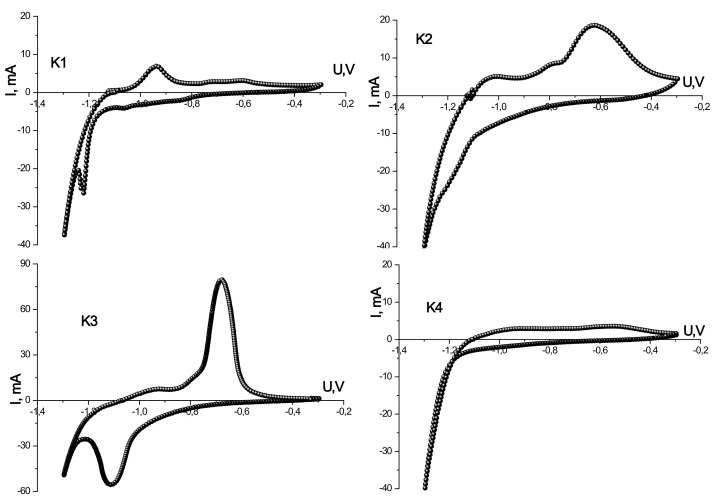
Cyclic voltammograms of the synthesized materials obtained at a scanning speed of 1 mV/s for samples K1, K2, K3, and K4.

**Figure 7 materials-14-06977-f007:**
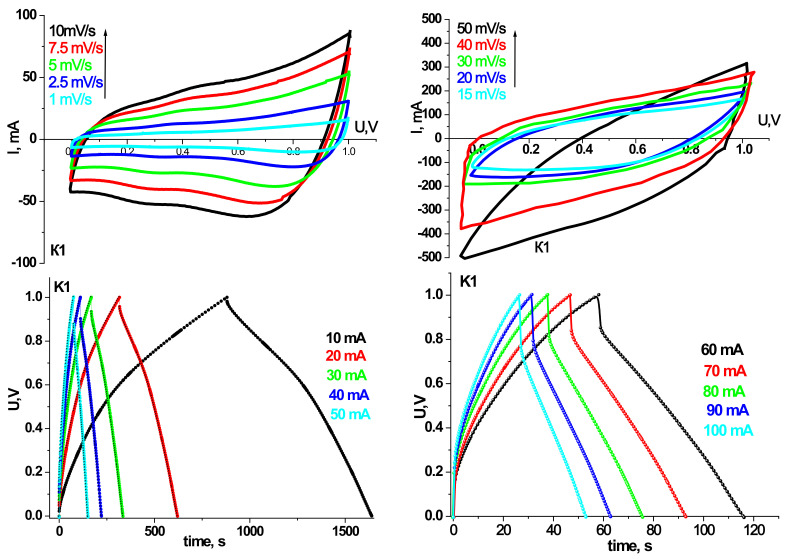
CVA and charge/discharge curves of the layout of HSCs based on the material K1.

**Figure 8 materials-14-06977-f008:**
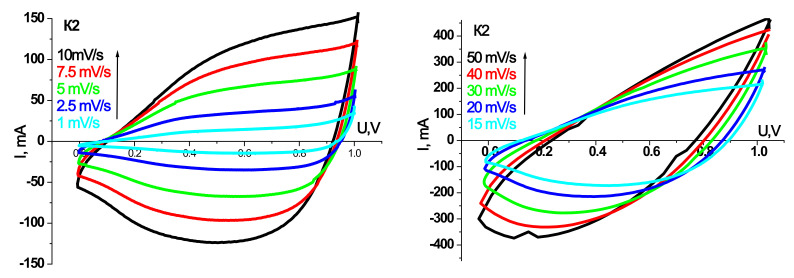
CVA and charge/discharge curves of the layout of HSCs based on the material K2.

**Figure 9 materials-14-06977-f009:**
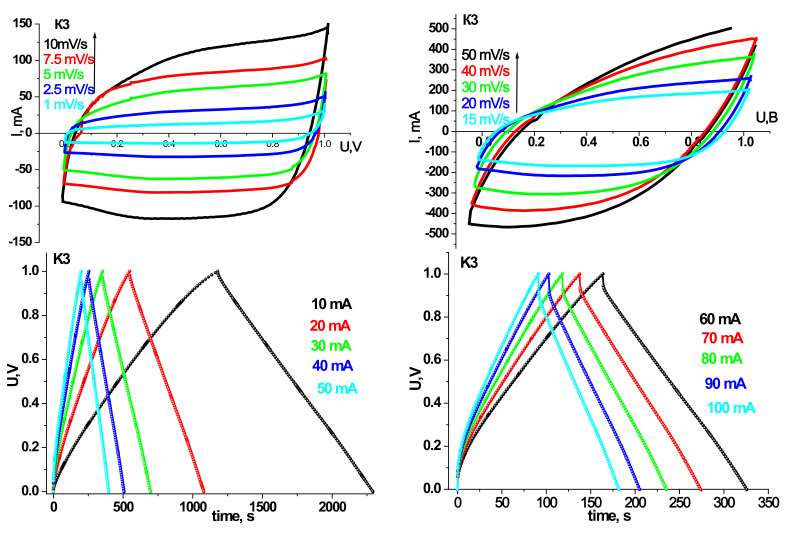
CVA and charge/discharge curves of the layout of HSCs based on the material K3.

**Figure 10 materials-14-06977-f010:**
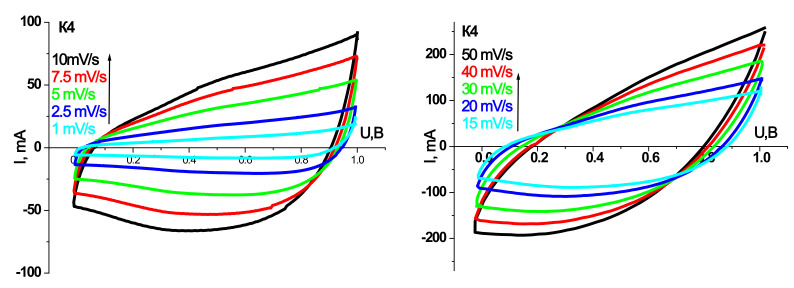
CVA and charge/discharge curves of the layout of HSCs based on the material K4.

**Table 1 materials-14-06977-t001:** The parameters of the spectra of samples obtained at different annealing temperatures.

No	Phase	δ, mm/s	Δ, mm/s	ω, mm/s	H_ef_, kOe	S, %
K1	Fe(OH)_3_	0.40	0.50	0.33	–	14
		0.39	0.81	0.41	–	39
		0.39	1.16	0.54	–	47
K2	Fe(OH)_3_/Fe(OH)_2_	0.98	2.60	0.74	–	11
		0.39	0.74	0.42	–	24
		0.38	1.11	0.62	–	65
K3	γ-Fe_2_O_3_/α-Fe_2_O_3_	0.37	−0.2	0.24	510	11
		0.32	−0.01	0.59	485	23
		0.36	−0.04	1.80	421	20
		0.33	0.67	0.34	–	9
		0.33	1.02	0.75	–	37
K4	γ-Fe_2_O_3_/α-Fe_2_O_3_	0.37	−0.21	0.26	514	35
		0.37	−0.18	0.29	501	5
		0.32	−0.02	0.68	485	26
		0.39	−0.01	1.02	424	8
		1.24	2.74	2.28	399	17
		0.31	0.93	0.80	–	9

Note: δ—isomeric shift, Δ—quadrupole splitting, ω—line width, S—integral intensity.

**Table 2 materials-14-06977-t002:** Calculated specific values of charging and discharging capacities and Coulomb efficiency for materials synthesized at different annealing temperatures.

No	T_anneal_ of Material	C_discharge_, F/g	C_charge_, F/g	Q_eff_, %
K1	Initial material	53	69	77
K2	150 °C	102	157	65
K3	250 °C	177	208	85
K4	350 °C	35	59	59

**Table 3 materials-14-06977-t003:** The specific capacity of HSCs on synthesized materials obtained from cyclic voltammograms.

Research Material		Sample K1		Sample K2		Sample K3		Sample K4	
No, n/N	S, mV/s	C_charge_, F/g	C_discharge_, F/g	C_charge_, F/g	C_discharge_, F/g	C_charge_, F/g	C_discharge_, F/g	C_charge_, F/g	C_discharge_, F/g
1	1	81	75	143	116	157	126	88	70
2	2.5	68	63	125	115	133	121	76	70
3	5	61	57	113	111	120	117	64	64
4	7.5	55	53	110	104	116	113	58	59
5	10	52	50	103	100	112	106	56	54
6	15	46	43	94	89	104	100	49	48
7	20	39	37	85	81	95	92	44	43
8	30	35	34	71	68	84	82	37	36
9	40	31	29	59	55	75	73	33	32
10	50	28	27	52	51	69	68	29	28

**Table 4 materials-14-06977-t004:** The specific capacity of HSCs on synthesized materials obtained from galvanostatic charge/discharge curves.

Research Material		Sample K1		Sample K2		Sample K3		Sample K4	
No, n/N	I, mA	C_charge_, F/g	C_discharge_, F/g	C_charge_, F/g	C_discharge_, F/g	C_charge_, F/g	C_discharge_, F/g	C_charge_, F/g	C_discharge_, F/g
1	10	93	81	158	134	133	124	85	73
2	20	67	65	132	123	124	122	70	67
3	30	54	53	119	109	121	120	64	62
4	40	48	47	113	111	118	117	59	58
5	50	41	40	107	106	116	114	56	54
6	60	38	37	106	102	114	112	53	51
7	70	35	34	99	98	111	110	50	48
8	80	34	33	96	95	107	106	47	46
9	90	31	30	92	91	105	104	44	43
10	100	29	28	91	89	103	101	42	41

## Data Availability

Data are contained within the article.

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
