# Peer review of "Structurally Dependent Electrochemical Properties of Ultrafine Superparamagnetic ‘Core/Shell’ γ-Fe2O3/Defective α-Fe2O3 Composites in Hybrid Supercapacitors"

_materials, 2021, doi:10.3390/ma14226977_

Round 1
Reviewer 1 Report
This manuscript presents a method for obtaining electrochemically active ultrafine composites of iron oxides, which involved modifying sol-gel citrate synthesis, hydrothermal treatment of the formed sol, and subsequent annealing of materials in the air. The electrochemical properties are studied for use in electrochemical capacitors. The different sections are clear and concise. I think that this paper is publishable in Materials. May be, authors should study the possibility of expressing the unit of conductivity using Siemens (S) instead of ohm-1.
Reviewer 2 Report
The manuscript is written cohesively and the results were rigorously discussed in comparison with the actual knowledge in the field. Please find below my point by point suggestions for improving this paper:
- I am reserved concerning the usage of core/shell concept in the title since this is only an assumption of the authors based on a proposed mechanism.
- The Abstract and Conclusion section could be shorter since the paper has an extensive Discussion part.
- Figure 3: please mark the K3 and K4 in the SEM images
- L216-217: “Either in the case of sample K1, which is a hydrogel of iron hydroxide Fe(OH)3 or for K2, which is a composite Fe(OH)3/Fe(OH)2“- It is unclear how the authors identified these phases for K1 and K2 samples since the XRD pattern did not present any diffraction maxima, as presented in L153-158.
- A general check-up and correction of the English language is suggested (e.g., L53-57 lack of verb, generally unclear; L 98 “an orange precipitate precipitated in the solution”; L236-237; L639-640 “there is the aggregation of superparamagnetic g-Fe2O3 particles are observed.)”
- The proposed subject seems to be outdated, based on the cited references (only 17% are from the 2017-2021 period).
Author Response
Thank you for your review and comments, which should improve our article.
The changes in the text of the manuscript are marked in bold or yellow.
Below please find the detailed response to all Reviewers' comments.
1. Regarding using the concept of "core/shell" to describe the structure of superparamagnetic particles of maghemite in our manuscript, we relied on the direct results of Mossbauer’s research and our experience of analyzing these results. In particular, in the case of synthesis of highly porous iron-containing materials under critical conditions of the reaction environment (auto combustion or hydrothermal treatment), there is a tendency to form a highly defective near-surface region of the material structure, in which some iron nuclei are in a different environment than inside the material structure. The Mossbauer effect clearly fixes this difference in the environment by quadrupole splitting of the component of the Mossbauer spectrum and analysis of their contribution to the total integral intensity [Synthesis, characterization and electrochemical properties of mesoporous maghemite γ-Fe2O3 , DOI:10.4028/www.scientific.net/SSP.230.120; The electrical conductivity and photocatalytic activity of ultrafine iron hydroxide/oxide systems, DOI: 10.1080/15421406.2018.1542070].
This type of approach, which consists in using Mossbauer studies to cross-confirm the different environments of iron nuclei and the formation of defective structures, in particular, and the core/shell structure is specified in other works [Metallorganic Routes to Nanoscale Iron and Titanium Oxide Particles Encapsulated in Mesoporous Alumina: Formation, Physical Properties, and Chemical Reactivity, DOI: 10.1002/1521-3765(20001201)6:23<4305::AID-CHEM4305>3.0.CO;2-N; Magnetic, X-ray and Mössbauer studies on Magnetite/Maghemite CoreShell Nanostructures Fabricated through Aqueous Route, DOI: 10.1039/b000000x;].
In our case, the superparamagnetic state of maghemite particles was noticed for sample K3, and there are two doublet components: with quadrupole splitting (D = 0.67 mm/s), which indicates the nuclei of superparamagnetic particles and large quadrupole splitting (D = 1.02 mm/s), which indicates iron nuclei in a highly ordered immediate environment. With large quadrupole fission, the last component was characterized as a highly defective material with a disordered environment of iron nuclei. Since the ratio of the integrated intensities is 9/37 towards the component with a larger quadrupole, it can be confirmed that the iron nuclei with a disordered environment are in the near-surface area superparamagnetic particles. This type of structure of superparamagnetic particles of maghemite was titled a core/shell. In the manuscript text, we described our arguments in more detail using the relevant references.
2. The Abstract and Conclusions were shortened while preserving their content.
3. It was made on the page.
4. Inaccuracy was admitted because the phases of materials K1 and K2 are not yet identified at this stage of the manuscript. This will be discussed below, namely L410-413 «For sample K1, the Mossbauer spectrum of which is a doublet line characteristic of 410 the paramagnetic state of ions Fe3+, the fixed values of isomeric shift (d) and quadrupole 411 splittings (D) are consistent with the parameters for the phase Fe(OH)3» та L431-432 «As a result of such transformations, a composite material of X-ray amorphous iron 432 hydroxides Fe(OH)3/Fe(OH)2 was formed».
This inaccuracy in the text was corrected.
5. The manuscript was carefully read and corrected for inaccuracies and mistakes. The comments made by the reviewer were considered.
6. In the references, we used authoritative and primary sources that are most relevant to the issues discussed in the manuscript and most deeply reveal them. Therefore, some of them were published earlier than in the period 2017-2021.
References to recent literature were added.
Reviewer 3 Report
The paper presents a method for obtaining electrochemically active ultrafine composites of iron oxides, superparamagnetic ‘core/shell’ -Fe2O3/defective α-Fe2O3, which involved modifying sol-gel citrate synthesis, hydrothermal treatment of the formed sol, and subsequent annealing of materials in the air. The synthesized materials' phase composition, magnetic microstructure, and structural, morphological characteristics have been determined via X-ray analysis, Mossbauer spectroscopy, scanning electron microscopy (SEM), and adsorption porometry. The mechanisms of phase stability were analyzed, and the model of phase transformations was suggested as FeOOH→-Fe2O3→α-Fe2O3. It was found that the presence of chelating agents in hydrothermal synthesis encapsulated the nucleus of the new phase in the reactor and interfered with the direct processes of recrystallization of the structure with the subsequent formation of the α- Fe2O3 crystalline phase. Additionally, the conductive properties of the synthesized materials were determined by impedance spectroscopy. The electrochemical activity of the synthesized materials was evaluated by the method of cyclic voltammetry using a three-electrode cell in a 3.5 M aqueous solution of KOH. For the ultrafine superparamagnetic ‘core/shell’ -Fe2O3/defective α-Fe2O composite with defective hematite structure and the presence of ultra-dispersed maghemite with particles in the superparamagnetic state was fixed increased electrochemical activity, and specific discharge capacity of the material is 177 F/g with a Coulomb efficiency of 85%. The prototypes of hybrid supercapacitor with work electrodes based on ultrafine composites superparamagnetic ‘core/shell’ -Fe2O3/defective α-Fe2O3 have a specific discharge capacity of 124 F/g with a Coulomb efficiency of 93% for current 10mA.
However the following issues should be addressed before it could be published.
Detailed Comments:
1.Key words too many, should highlight the key points。
2.In line 57, the last line, the colon should be inside the period, not outside.
3.The data format in Table 2 has a problem and needs to be modified again.
4.Table 2 has a formatting problem with the following text content and needs to be modified.
5.The small diagrams in FIG. 7 FIG. 8 and FIG. 10 are not aligned,They should be aligned for aesthetic reasons.

Author Response
Thank you for reviewing the article and the comments to it!
The changes in the text of the manuscript are marked in bold or red.
Below please find the detailed response to all Reviewers' comments.
- The number of keywords was reduced.
- The inaccuracy was corrected.
- It has been corrected in the manuscript.
- The comments were considered, and the table was replaced.
- All figures were adjusted to achieve a clear understanding and aesthetic feasibility.
Reviewer 4 Report
- I suggest changing some keywords as I could not find them more than 1 or two times in the manuscript. You are supposed to select keywords based on more repetition words in your research. Please replace new keywords with mesopores, hybrid supercapacitors.
- Section 2: materials are missing. Please bring the chemical name of all the materials that you used in this study coming with the brand and the country of the supplier (refer to the template of the journal).
- Figure 3. (K1) and (K2) are not mentioned in the figure. Please compare all in the text.
- Figure 4. Why Adsorption-desorption isotherms are not performed for all the samples?
- Table 2 is not performed properly in the manuscript.
- What is statistical analysis in this research?
- You may compare all your results in section 3 with other research studies. For example, in X-ray section you can use the following reference:
Sabbagh, F., Kiarostami, K., Mahmoudi Khatir, N., Rezania, S., & Muhamad, I. I. (2020). Green synthesis of Mg0. 99 Zn0. 01O nanoparticles for the fabrication of κ-Carrageenan/NaCMC hydrogel in order to deliver catechin. Polymers, 12(4), 861.
Author Response
Thank you for your review and comments. We will take into account all the reviews of the reviewer
- We agree with this remark, and it was changed the keywords.
- This remark was considered. Appropriate changes were made to the manuscript text
- It was changed the Fig.3.
- We did not specify the adsorption-desorption isotherms for samples K1 and K2 because they do not have a physical meaning to determine future structural and morphological characteristics. Due to the desorption of structurally bound water from materials K1 and K2 when measuring adsorption porometry, the corresponding desorption curves prevailed over adsorption, and no closed hysteresis was observed. Therefore, according to them, it was not possible to obtain any good result for analysis. We have added this question in the manuscript text. Additionally, we do not consider it appropriate to provide experimental data for samples K1 and K2, as we did not use them for analysis.
- It was changed on the page
- According to the manuscript's text, we did not use “statistical analysis” in the study. However, if we consider the method of Mossbauer spectroscopy used, it is based on statistical bases because the corresponding spectra result from resonant absorption of gamma quanta on iron nuclei in the structure of the material. This process is statistical; as a result, it is obtained based on a statistical data set of gamma-ray counters.
- It was added on the page
Reviewer 5 Report
In this manuscript entitled “Structurally dependent electrochemical properties of ultrafine superparamagnetic ‘core/shell’ gamma-Fe2O3/defective alpha-Fe2O3 composites in hybrid supercapacitors”, the authors performed comprehensive studies on the effects of post-annealing temperature on the structural, electrical, and electrochemical properties of the Fe2O3 materials. The topic of this manuscript is of the interests of the community and fits materials journal well. The design of the experiment and the analysis of results are sounds. I therefore recommend this manuscript to be published in materials.
Author Response
Thank you for your review and your time!